# Noninvasive Electromagnetic Neuromodulation of the Central and Peripheral Nervous System for Upper-Limb Motor Strength and Functionality in Individuals with Cervical Spinal Cord Injury: A Systematic Review and Meta-Analysis

**DOI:** 10.3390/s24144695

**Published:** 2024-07-19

**Authors:** Loreto García-Alén, Aina Ros-Alsina, Laura Sistach-Bosch, Mark Wright, Hatice Kumru

**Affiliations:** 1Fundación Institut Guttmann, Institut Universitari de Neurorrehabilitació Adscrit a la UAB, 08916 Badalona, Spain; superaina@gmail.com (A.R.-A.); laurasistachbosch@gmail.com (L.S.-B.); mawright@guttmann.com (M.W.); 2Universitat Autónoma de Barcelona, 08193 Barcelona, Spain; 3Fundació Institut d’Investigació en Ciéncies de la Salut Germans Trias i Pujol, 08916 Badalona, Spain

**Keywords:** electromagnetic stimulation, noninvasive neuromodulation, functionality, motor function, upper limb, cervical spinal cord injury

## Abstract

(1) Background: Restoring arm and hand function is one of the priorities of people with cervical spinal cord injury (cSCI). Noninvasive electromagnetic neuromodulation is a current approach that aims to improve upper-limb function in individuals with SCI. The aim of this study is to review updated information on the different applications of noninvasive electromagnetic neuromodulation techniques that focus on restoring upper-limb functionality and motor function in people with cSCI. (2) Methods: The Preferred Reporting Items for Systematic Reviews and Meta-Analysis (PRISMA) guidelines were used to structure the search protocol. A systematic review of the literature was performed in three databases: the Cochrane Library, PubMed, and Physiotherapy Evidence Database (PEDro). (3) Results: Twenty-five studies were included: four were on transcranial magnetic stimulation (TMS), four on transcranial direct current stimulation (tDCS), two on transcutaneous spinal cord stimulation (tSCS), ten on functional electrical stimulation (FES), four on transcutaneous electrical nerve stimulation (TENS), and one on neuromuscular stimulation (NMS). The meta-analysis could not be completed due to a lack of common motor or functional evaluations. Finally, we realized a narrative review of the results, which reported that noninvasive electromagnetic neuromodulation combined with rehabilitation at the cerebral or spinal cord level significantly improved upper-limb functionality and motor function in cSCI subjects. Results were significant compared with the control group when tSCS, FES, TENS, and NMS was applied. (4) Conclusions: To perform a meta-analysis and contribute to more evidence, randomized controlled trials with standardized outcome measures for the upper extremities in cSCI are needed, even though significant improvement was reported in each non-invasive electromagnetic neuromodulation study.

## 1. Introduction

Loss of arm and hand function is one of the most devastating consequences in people affected by a cervical spinal cord injury (cSCI), and it has been shown to be the priority of recovery for this population [1,2,3,4]. The degree of impairment depends on the level and severity of injury [1] and typically results in reduced independence in the performance of activities of daily living and limited participation [5]. Spinal cord injury (SCI) disrupts communication between the brain and body and induces an interruption of the neural pathway that controls movement [6]. Additionally, signal transduction, axonal growth and myelination are disrupted, inhibiting the recovery of spinal cord function [6]. Therefore, one of the goals for restoring function after SCI is neural circuit reconstruction to achieve recanalization of the neural pathway [7]. Over the last few decades, various interventions, such as functional training, tendon transfer surgery, implanted neuroprostheses, and neuromodulation, have evolved in an attempt to improve upper-limb function in individuals with SCI [1]. Neuromodulation has been the fastest-growing discipline in the field of medical sciences [7]. This intervention achieves therapeutic effects by altering the function or state of the nervous system via invasive or noninvasive electromagnetic stimulation [7]. This review focused on noninvasive techniques due to greater accessibility to the clinical environment. Noninvasive electromagnetic neuromodulation can be achieved through different types of stimulation that can be classified depending on which part of the nervous system is acted upon. Central nervous system stimulation includes repetitive transcranial magnetic stimulation (rTMS), transcranial direct current stimulation (tDCS) and transcutaneous spinal cord stimulation (tSCS) [7,8,9,10]. On the other hand, functional electrical stimulation (FES), transcutaneous nerve stimulation (TENS), and neuromuscular stimulation (NMES) are the three main forms of peripheral nervous system stimulation [7,8,9,10].

rTMS is a cortical stimulation technique that uses a magnetic coil to deliver a series of magnetic pulses to the brain, which can modulate the activity of neurons [9,10,11]. Depending on the parameters of the stimulation, rTMS can either enhance or inhibit the activity of the targeted brain area. rTMS is typically applied to the scalp over the target area, and the stimulation can be delivered in a single session or over a series of sessions. The most common rTMS protocols involve high-frequency stimulation (5–20 Hz) for facilitation and low-frequency stimulation (1 Hz) for inhibition [9,10,11].

tDCS involves applying a weak electrical current (usually 1–2 milliamps) to the scalp to stimulate or inhibit specific areas of the brain [8,9,10,11]. Current flows from an anode (positive electrode) to a cathode (negative electrode) and is typically applied for 20–30 min. The exact mechanism of action of tDCS is not completely understood, but it is thought to modify the excitability of neurons in the targeted area of the brain. Motor training usually promotes activity-dependent plasticity [8,9,10,11].

tSCS is a method for potentially activating spinal cord circuitry through electrodes placed on the skin over the vertebral column [9,11,12]. The objective is to activate residual neural networks in the spinal cord that are inaccessible after SCI through the tonic activation of afferent fibres from the posterior roots of the spinal cord [13,14,15,16,17].

FES is a technique that uses bursts of short electric pulses (pulse width 0–250 ms and amplitude 10–150 mA) to generate muscle contraction by stimulating muscles with the aim of restoring or improving specific functional abilities in individuals with neurological or musculoskeletal impairments [18,19]. Simultaneously, a number of muscle groups are stimulated to coordinate the movement of functional activities such as grasping, releasing, and walking [18,20,21]. The key element for achieving synergistic activity in muscles is the appropriate sequencing of bursts of electrical pulses [19]. The most common current application uses surface electrodes in the vicinity of the motor point of the muscles involved in functional movement of the upper limb to restore hand functionality [18,20].

TENS is applied through surface electrodes on the skin, which activate nerves through low frequencies (<10 Hz) to produce muscle contraction or high frequencies (>50 Hz) to produce paraesthesia without muscle contraction [21]. TENS is a treatment that is traditionally used for pain management, but recent studies have shown that it can improve hand motor function and performance [21].

NMES is a technique that generates muscle contraction by creating an electrical field near motor axons of peripheral nerves, which depolarizes the axonal membranes and stimulates action potentials, leading to muscle contractions [21]. This technique can be applied transcutaneously with surface electrodes positioned over the target muscles, percutaneously with intramuscular electrodes that are connected to an external simulator, or subcutaneously with an implanted simulator [21]. NMES can restore motor function in individuals who have muscle weakness or paralysis. When combined with functional task practice, it is thought to improve recovery by promoting adaptive neuroplasticity [21].

The noninvasive nature, affordability, and clinical accessibility of these techniques make them attractive treatment options for promoting motor and functional recovery of the upper extremities in individuals following cSCI. Although neuromodulation techniques have been found to be useful for preventing neuronal dysfunction following SCI, the mechanisms of neural circuit reconstruction remain unknown, and there are a wide variety of intervention protocols. The aim of this systematic review is to provide updated information on the different application protocols, measurement of results, adverse effects and therapeutic effects of noninvasive electromagnetic neuromodulation techniques for restoring upper-limb functionality and motor function in people with cSCI.

## 2. Materials and Methods

### 2.1. Search Strategy

The Preferred Reporting Items for Systematic Reviews and Meta-Analysis (PRISMA) guidelines were used to structure the search protocol. Four authors reviewed the titles and abstracts, excluding studies that did not meet the established exclusion criteria, before reading the full texts of the selected articles. The search was conducted between September and December 2023 using the Cochrane Library, PubMed, and Physiotherapy Evidence Database (PEDro) electronic databases. To obtain the required information, we used the medical subject headings (MeSH) terms “spinal cord injuries”, “upper limb”, “transcranial magnetic stimulation”, “transcranial direct current stimulation”, “spinal cord stimulation”, and “transcutaneous electrical nerve stimulation”, as well as the free terms “functional electrical stimulation”, “neuromuscular stimulation”, “transcutaneous spinal cord stimulation” and “sensory stimulation” combined with Boolean operators “AND” and “OR”. The search was limited to English- and Spanish-language human studies.

### 2.2. Study Selection Procedure

Using the PICOS structure, we established the following inclusion criteria: (i) more than 18 years old with cSCI; (ii) noninvasive electrical or magnetic neuromodulation of central or peripheral nervous system intervention; (iii) evaluation of motor function and/or functionality of the upper limb; and (iv) randomized controlled clinical trials or those with a crossover design. Articles with the following criteria were excluded: (i) intervention applied that combined two or more types of stimulation and (ii) intervention applied after a tendon and nerve transfer.

### 2.3. Assessment of Methodological Quality

The Physiotherapy Evidence Database (PEDro) Scale was used to assess the methodological quality of the included studies [22]. There are a total of 11 items that are answered with a “yes” (score = 1) or “no” (score = 0). Item 1 refers to the external validity of the study and is not used to calculate the final score. Items 2–9 refer to internal validity, with items 10 and 11 indicating whether the statistical information provided by the authors allows for an adequate interpretation of the results. The total score ranges from 0–10 points, in which higher scores represent better methodological quality: high quality, a score equal to greater than 7; moderate quality, a score of 5–6; and poor quality, a score of 4 or less.

The scores were obtained from the PEDro website for all trials, except ten, which were scored by the authors because they were not specified.

### 2.4. Data Extraction and Statistical Analysis

All the data were extracted from the studies included by four different authors. The following descriptive data were extracted: authors, publication year, study design, number of subjects in each experimental group, subject clinical and demographic characteristics, intervention characteristics (type, frequency and duration of the sessions, stimulation parameters and total duration of the intervention), and outcome measures related to the functionality and motor function of the upper extremities. Meta-analysis calculations were performed using Review Manager Software (version 5.4). Motor function and functionality outcomes were extracted from each study to determine the mean and standard deviation of the change in post-intervention scores, adjusted for the baseline score for each group (95% confidence interval). The data extracted were expressed as the mean difference.

Statistical heterogeneity was quantified using the I^2^ statistic, where an I^2^ > 75% was considered to indicate excessive heterogeneity. A fixed-effects model was used to pool data if the Iº was less than 50%, and a random-effects model was used if the I^1^ was between 50 and 75%. Effect sizes were interpreted using Cohen’s guidelines (0.2 = small effect, 0.5 = moderate effect, 0.8 = large effect, >0.8 = very large effect).

## 3. Results

### 3.1. Study Selection and Characteristics

A total of 246 eligible studies were identified after searching the databases, and 10 from other sources.

After eliminating 31 duplicates and 225 articles screened, 28 were chosen for full-text screening, and 25 were included in the systematic review. The study selection process is shown in Figure 1.

Among the 25 included studies, 7 (28%) were crossover studies, and 18 (72%) were randomized controlled trials. Four studies focused on TMS, 4 on tDSC, 2 on tSCS, 10 on FES, 4 on TENS, and 1 on NMS. The study design and subject characteristics of the included studies are detailed in Table 1.

### 3.2. Effect of Interventions

The intervention characteristics of the included studies are detailed in Table 2.

### 3.3. Transcranial Magnetic Stimulation (TMS)

Four crossover studies reported the effect of TMS for upper limbs (ULs) [23,24,25,26]. The sample sizes were 4 and 15 subjects. The subjects were between 26–70 years old and affected by a cSCI with AIS of A to D at C2 to C8, and 3 to 343 months since injury. From 3 to 10 sessions were applied with a frequency of 5 days per week. Only the Kuppusway et al. and Belci et al. studies specified the sessions’ durations, 15 min and 1 h, respectively. The four studies compared the effects of active and sham conditions. Sham conditions involved rotating the coil 90° to ensure no brain stimulation [23], delivering only 5% of the real stimulator output [24], or positioning the coil over the occipital cortex [25]. All the studies applied the stimulation over the hand representation in the motorocortex [23] (lowest threshold spot in FDI, ECR or thenar eminence [24]; left motorcortex [25], or hemisphere contralateral to the weaker hand [26]. rTMS was delivered differently among the studies: 3 stimuli in 200 ms with intervals of 2 s at 50 Hz [23]; 2 trains with intervals of 8 s for 15 min at 5 Hz [24]; double pulses separated by 100 ms (10 Hz) at a frequency of 0.1 Hz (10 s interval) [25], or 800 pulses distributed in trains of 40 pulses in 2 sec with an inter-train of 30 s at 19 Hz [26]. Intensity was adjusted to 80% of the resting motor threshold (RMT) of the ULs [23], at the active motor threshold (AMT) of the FDI, ECR or thenar eminence [24], 90% MEPs in the hand muscles [25], or at 80% RMT in the biceps [26]. Only one study reported headache after intervention in one subject [26].

Upper-limb (UL) motor function was evaluated by the Upper Extremity Motor Score (UEMS), except Gomes-Osman and Field-Fote, who only evaluated pinch and grip force. Evaluation of functional outcomes varied among the studies. Gharroni et al. evaluated independence in ADLs by the SCIM [23], whilst the other studies evaluated UL functionality by the Action Research Arm Test (ARAT) [24], Nine Hole Peg Test (NHPT) [24,25], and JTHFT [26].

For motor function, only one study showed significantly improvements in UEMS score [25] after active rTMS, but the results were not significant when compared with sham group. Gomes-Osman and Field-Fote demonstrated significant improvements in grasp, but no differences when compared with the sham group [26]. With respect to functionality, significant improvements were demonstrated in the ARAT and JTHFT tests after active rTMs but no significant differences were found compared to the sham group. Finally, Gharroni et al. showed no significant differences in the SCIM score in both groups [23].

The evaluation of the methodological quality of the studies revealed moderate quality, with an average of 6.5 points on the PEDro scale. The meta-analysis could not be completed because of the lack of motor function data [24,25] and a lack of common evaluation functional tests [23,24,25,26].

### 3.4. Transcraneal Direct Cortical Stimulation (tDCS)

Two RCTs and two crossover studies reported the effect of tDCS on UL [27,28,29,30]. The sample sizes of the crossover studies were 11 and 9 subjects, respectively, while the RCT sample sizes were 4 subjects in each group. The subjects ranged from 20 to 63 years old; had a cSCI with AIS of B, C or D at levels C3 to C7; and were 7 to 372 months since injury. One study combined tDCS with robotic-assisted training [29], another combined tDCS with massive amounts of practice [27], and the other two applied tDCS alone [29,30]. The number of sessions ranged from 3 to 10, with a duration of 20 min per session, and the sessions were distributed between 1 and 5 sessions per week. The localization of the anode electrode was C3/C4 in the M1 region contralateral to the target arm, except for one study, in which the anode electrode was applied in the weakest muscle below the level of injury [27]. The cathode electrode was placed over the contralateral supraorbital area. The intensity was set at 2 mA, and two studies added one additional condition with 1 mA [29,30]. Sham conditions were performed with a short ramp-up/ramp-down event at the beginning and end of the stimulation period without any current between the two events. Murray et al. and Yozbatiran et al. specified tingling, skin redness, sleepiness [28,30], headache and itching under the electrode as adverse effects [30]. All these symptoms disappeared soon after the cessation of the intervention and ranged from mild to moderate in severity.

There was no standardization in the outcome measures used, with studies using different scales for UL functionality: box and block test (BBT), motor activity log (MAL), Jebsen–Taylor hand function test (JTHFT), ARAT, and NHPT. For UL motor function, the maximal voluntary contraction (MVC) test and manual muscle test (MMT) were used.

Only one study showed significant improvements in UL functionality compared with the sham group [28]. Potter-Baker et al. showed significant improvements in both groups for UEMS, ARAT, and NHPT, without difference between groups [27]. Finally, Murray et al. and Cortes et al. did not report any change for MVC and BBT in any condition, respectively [29,30].

The evaluation of the methodological quality of the studies revealed moderate quality, with an average of 6.5 points on the PEDro scale. The meta-analysis could not be completed because of the lack of motor function information [27,28] and a lack of common evaluations of functional tests [27,28,29].

### 3.5. Transcutaneous Spinal Cord Stimulation (tSCS)

Only two crossover studies have investigated the effect of tSCSs on UL [31,32]. Between 6 and 21 cSCI subjects were studied, ranging from 18 to 70 years old, with AIS A/B/C/D at C3 to T1. The time since injury was between 1.5 and 12 years in one study [32] and between 3 to 10 months in the other study [31]. In both studies, tSCS combined with intensive functional task training and robotic exoskeletons was applied in the intervention group. tSCS was applied by two cathode electrodes at and below the injury level [32] or at C3-C4 and C6-C7 [31], and two anode electrodes over the iliac crest. tSCS was applied by a biphasic or monophasic waveform with a 1 ms pulse and a frequency of 30 Hz (10 kHz). The intensity was adjusted according to the facilitation of manipulation [32] or at 90% of the threshold of abductor pollicis brevis (APB) [31]. Some adverse effects were reported during tSCS, such as worsening of spasticity in the upper limb, nausea, cough, increased tingling sensation in the lower limb, and dysreflexia during stimulation, which ceased when tSCS was stopped [31].

GRASSP, SCIM III, and BBT were used to evaluate functionality. To evaluate motor function UEMS grip and pinch force were used.

The results demonstrated that a combination of tSCS and UL training significantly improved UL strength, prehension ability, and pinch force when compared with UL training alone [31,32]. Also, all the subjects improved by up to 8 points in the UEMS score at the end of tSCS with UL training compared to 2 points or fewer following UL training alone [32]. In the case of SCIM, the self-care domain improved by 1 to 4 points for each participant following tSCS and UL training [32]. García-Alén et al. demonstrated that only the intervention group improved significantly in lateral pinch force, but the change in score was not significant between the groups [31].

The evaluation of the methodological quality of the studies revealed moderate quality, with an average of 4.5 points on the PEDro scale. The meta-analysis could not be completed because only two RCTs were included.

### 3.6. Functional Electric Stimulation (FES)

Ten RCTs investigated the effect of FES on UL [33,34,35,36,37,38,39,40,41,42]. The control groups included 3 to 33 subjects, and the intervention groups included 3 to 37 subjects. The age of the subjects who participated ranged from 22 to 63 years, and they were affected by a cSCI with AIS of A to D at C3-C7 and from 2 weeks to 16 months since injury. FES was applied to the wrist muscles [34,35,36,38,39,41], flexor and extensor digitorium, and thumb flexor abductors and oppositors [38,39,41]. Two studies applied surface electrodes in the muscle that facilitated grasp and/or pinch performance [37,42]. The parameters varied between studies that applied pulse widths of 200 or 250 μs, except for two studies that used 0.3 ms [35,36]. Only two studies mentioned the criteria established to adjust the intensity: being set at an appropriate level of tolerance [35,36]. The intensity amplitude ranged from 8 to 70 mA, and three studies did not specify it [34,36,42]. The stimulation was applied at frequencies ranging from 20 to 70 Hz. Anderson et al. used pre-programmed stimulation protocols [33].

FES has been combined with robotic therapy [34], resistance training [35], hand training [33,37], functional patterns [38,42], conventional occupational therapy (COT) [39,41], biofeedback [36], and ADLs [40]. Only one study compared active with sham FES stimulation [35], whilst the other studies compared active with conventional occupational therapy (COT). The total number of sessions ranged from 24 to 84, with a frequency of 3 or 5 sessions per week and a duration ranging from 20–40 min to 2 h. Only one study reported adverse effects in one participant, who experienced redness at the electrode site, left dorsum hand swelling, and right ventral forearm swelling [33].

For motor function, the MVC and MMT were used for evaluation. On the other hand, for functionality, the ARAT and GRASSP were used, whilst four self-feeding abilities were graded through four tasks—FIM, SCIM, TRI-HFT, REL test, and CUE.

Only two studies evaluated UL motor function and showed no benefits when applying FES combined with progressive resistance exercises in the wrist or with biofeedback [35,36]. For UL functionality, the authors reported significant improvements when combining FES with biofeedback [36], hand training [33,37], COT [41], functional tasks [42] and robotic training [34]. When compared with a control group, only two studies, which applied FES combined with COT or ADLs, showed significant results [39,40]. Finally, one study did not perform statistical analysis due to the small sample [38].

The evaluation of the methodological quality of the studies revealed moderate quality, with an average of 5.1 points on the PEDro scale. Meta-analysis for motor tests could not be completed due to a lack of the common evaluations for motor and functional tests.

### 3.7. Transcutaneous Electrical Nerve Stimulation (TENS)

Four RCTs reported the effect of TENS on the UL [43,44,45,46]. The sizes of the intervention group ranged from 5 to 14 subjects, while the sizes of the control group ranged from 5 to 10 subjects. The subjects were 22 to 70 years old and had a cSCI with AIS of A to D at C4-C8 and were 7 months to 13 years since injury. Three studies applied two or three conditions in the intervention group: massive practice, TENS, or the combination of both [44,45,46]. The number of sessions ranged from 15 to 17, with a 2 h duration per session, distributed in 5 sessions per week. TENS was applied through the anode electrode at the wrist, with the cathode electrode at a distance of 2 cm from the anode. The intensity was adjusted to muscle action potentials from APB [43,45,46] or to the point when an observable twitch was evoked in any of the muscles innervated by the median nerve [44]. No study reported adverse effects.

To evaluate motor function, AIS and grasp and pinch grip force were used. For UL functionality, the JHFT and the WMFT were used.

TENS combined with massive practice (MP) showed significant improvements in UL functionality and pinch grip force compared with the control group [43,44,45], and both techniques were applied alone [43,44]. Only one study evaluated UL motor function, demonstrating that TENS combined with MP significantly improved UEMS when compared with MP alone and conventional therapy [45]. Gomes-Osman and Field-Fote showed significant improvements in precision grip after intervention of TENS combined with FTP, but the results were not significant when compared with those of the control group [46].

The evaluation of the methodological quality of the studies revealed moderate quality, with an average of 4.3 points on the PEDro scale. Meta-analysis for motor function could not be completed since only one study evaluated UEMS [45]. Despite the standardization in the assessment of UL functionality and pinch grip force, meta-analysis could not be completed due to a lack of data [43,46].

### 3.8. Neuromuscular Stimulation (NMS)

In the literature, only one neuromuscular stimulation study included RCT conditions for the upper limb in cSCI patients [47]. The intervention group consisted of two groups: one group that received 8 weeks of NMS assisted by arm ergometry exercise, and the other group that received four weeks of NMS assisted by exercise followed by four weeks of voluntary arm crank exercise. The control group only performed voluntary exercise for 8 weeks. A total of 23 subjects participated in the intervention, 12 in one condition and 11 in the other; 12 subjects participated in the control group. The age range was between 18 and 45 years, the level of injury and AIS were not specified, but the time since injury ranged from 4 to 9 years. There were 24 total sessions, which were distributed in 3 sessions per week. NMS was applied through an electrode near the motor point of the triceps and another placed 5 cm distal to the first electrode, along the long axis of the humerus. NMS was applied at a frequency of 50 Hz with a pulse width of 250 μm for 10 min. The intensity was adjusted to produce optimal contraction of the triceps muscle. There were no adverse effects in any of the subjects who participated.

The subjects who performed NMS combined with arm ergometry exercise had significantly greater UL strength than the subjects who voluntarily exercised without NMS.

The evaluation of the methodological quality of the studies revealed poor quality, with only 2 points on the PEDro scale. The meta-analysis could not be completed because there was only one RCT focused on NMS for UL.

## 4. Discussion

This systematic review investigated the effects of different noninvasive electromagnetic neuromodulation techniques, with or without rehabilitation for improving the motor function and functionality of ULs in subjects with cSCI. Unfortunately, the meta-analysis could not be completed due to a lack of common motor or functional evaluations across studies. Also, there was big variability in the evaluation of muscle strength and/or functionally of UE in SCI and the expression of those data; for example, one study gave only percentage changes without any rough data or mean +/− SD [5]. There should be a general concept for clinical assessments for UE, such as UEMS of ASIA for muscle strength and GRASSP for functionally assessment, and the data should be expressed as rough data and group mean. For each technique, we could only include a few studies, between 1 and 4, except for FES, which had 10 studies included. The strength of evidence was not high for any of these studies; moderate for the studies focused on rTMS, tDCS, tSCS, FES, and TENS; and poor for the only study that applied NMS. Apart from four studies [31,33,35,42], the sample sizes were small, oscillating from 3 to 15 subjects in each experimental group, limiting the statistical power and reliability of the findings. It is necessary to encourage multicenter trials and collaboration to increase sample sizes. Also, the samples were highly heterogeneous regarding the severity of the SCI, time since injury, and age, impacting the power and generalizability of the results. The conducting of subgroup analyses or stratified meta-analysis based on theses variables can provide nuanced insights. Future studies should aim for more homogenous sample populations, or explicitly address and adjust for these viabilities in their analyses.

Furthermore, there is a lack of information in some studies related to the time since injury and the subject’s age [34,38].

Similarly, therapy protocols were not standardized among the studies included in the present review. The types of training included conventional occupational therapy, robot-assisted arm training, massive practice, hand training, resistance training, biofeedback, repetitive grasping exercises, and ergometers. Also, the parameters of stimulation (intensity settings, frequency, pulse width, stimulation points) and intervention characteristics (type, number of sessions, frequency, and duration) varied. All these factors may play a significant role in the effectiveness of UL recovery on motor function and functionality, so it is important to standardize UL recovery to achieve high-quality evidence. Therefore, these different possibilities for intervention should be considered carefully, and further studies are necessary to provide reliable information for clinical application. Also, there is a lack of information in some studies related to session frequency [26,28,29,30] and duration [23,26,34,35,47] and the criteria to adjust the intensity of the stimulation [34,37,38,39,40,41,42].

Only TMS and tDCS studies and one FES study applied sham stimulation in the control group; therefore, most of the studies could not perform blinded interventions, subtracting from the study quality [23,24,25,26,27,28,29,30,35].

The reporting of adverse effects is inconsistent, with some studies not mentioning them at all. A comprehensive reporting of adverse effects in future studies would be important to better understand the safety profile of each neuromodulation technique.

Nonetheless, the outcomes obtained from TMS studies showed improvements in UL functionality, but the improvements were not significant compared with those in the sham group [24,26]. tDCS combined with rehabilitation improved UL functionality [28] and UL motor function [27], but the difference was not significant compared with that in the sham group. tSCS combined with rehabilitation significantly improved UL functionality compared with rehabilitation alone [31,32].

Furthermore, outcomes obtained from peripheral neuromodulation studies have shown that the combination with rehabilitation significantly improves UL functionality [33,34,36,37,39,40,41,42,43,44,45], pinch force [43,44,45,46], and UL motor function [45,47]. For UL functionality, the results compared with the control group were significantly in two FES studies [39,40] and three TENS studies [43,44,45]. On the other hand, for UL motor function, results compared with the control group were significant in one TENS study [45] and one NMS study [47]. Two studies showed improvements in UL motor function and functionality, and pinch force when applied TENS alone, but the results where highly significant when combined with rehabilitation [44,46]. Future studies should include longer follow-up periods to assess the durability of the therapeutic effects.

The field of electromagnetic neuromodulation is developing rapidly, but studies carried out to date are still limited.

### Limitations

In this systematic review, only English- and Spanish-language articles were included because some published studies in other languages were missing. Because this kind of technique is a new research field, there are few publications in the literature, and the quality is low. The small sample size negatively influences the power of the studies to detect an effect. Meta-analysis was not possible because of a lack of common assessments and standardization of the data presentation, such as mean and standard deviation. Additionally, the heterogeneity of the sample, time since SCI, and the results of the inter-individual variability in response to the intervention may have influenced the magnitude of the upper-limb functionality and motor function.

## 5. Conclusions

In order to perform a meta-analysis, more randomized controlled trials with standardized outcome measures for the UL in cSCI are needed. Future research should prioritize the use of standardized outcome measures to facilitate meta-analysis. Authors should be encouraged to adopt commonly accepted scales such as UEM for UL motor function and GRASSP for UL functionality in cSCI. Other additional clinical assessment could be done, such as UEMS, ARAT, and NHPT, and the data should be given adequately, such as rough data, mean, and SD. Some studies have low methodological quality, as indicated by low PEDro scores. Future research should emphasize the importance of high-quality study designs, such as randomized controlled trials, with clear reporting and adherence to established methodological guidelines like the CONSORT statement. Better knowledge about the effectiveness of noninvasive electromagnetic neuromodulation techniques can help clinicians to use it safely and effectively in their clinical environment.

## Figures and Tables

**Figure 1 sensors-24-04695-f001:**
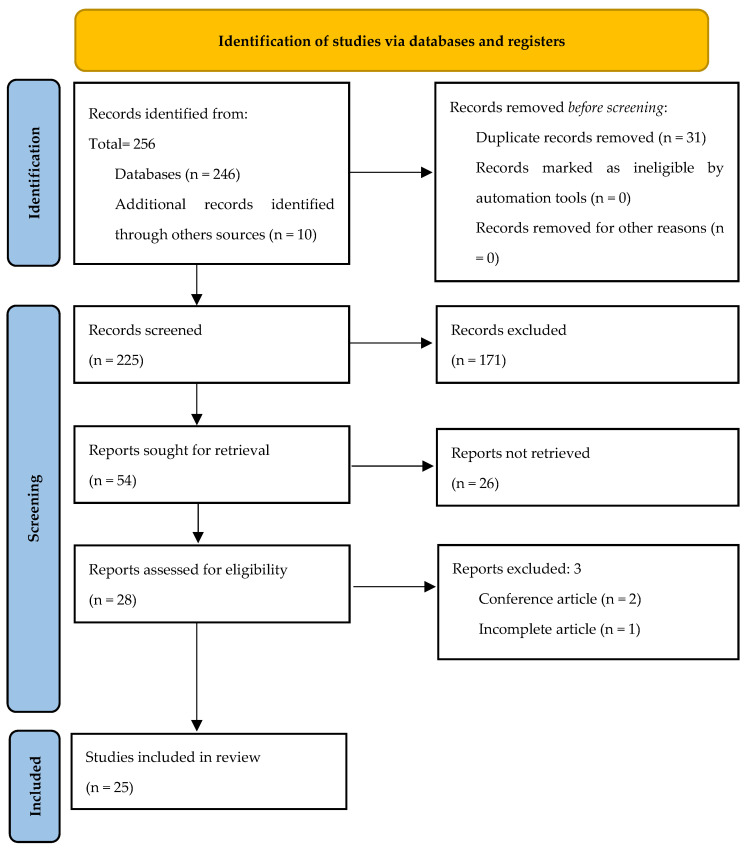
PRISMA flow diagram of the systematic review process.

**Table 1 sensors-24-04695-t001:** Main characteristics of the studies included in the systematic review.

Study	Design	Intervention Groups	N° Subjects per Group	Level and AIS	Time since Injury Mean (SD) and/or Range	Age Mean (s.d) or Range	PeDro Scale
**TMS (Transcranial Magnetic Stimulation)**							
Gharooni et al., 2018 [23]	Crossover	EG: active iTBS CG: sham iTBS	10	C3-C6, AIS B-D	11.40 (14.96) mo, range = 3–46 mo	46.80 (12.50) yrs, range = 29–70 yrs	6
Kuppuswamy et al., 2011 [24]	Crossover	EG: active rTMS CG: sham rTMS	15	C2, C4-C8, AIS A-D	116.7 (90.3) mo, range = 3–343 mo	39.7 (9.8) yrs, range = 26–59 yrs	5
Gomes-Osman and Field-Fote, 2015 [25]	Crossover	EG: active rTMS+RTP CG: sham rTMS+RTP	11	C6 (median), AIS C-D	6.6 (8.2) yrs, range = 1–14 yrs	46.7 (12.0) yrs, range = 34–58 yrs	9
Belci et al., 2004 [26]	Crossover	EG: active rTMS CG: sham rTMS	4	C5, AIS D	5.7 (3.2) yrs, range = 1–8 yrs	43.7 (13.3) yrs, range = 26–54 yrs	6
**TDCS (Transcraneal Direct Current Stimulation)**							
Yozbatiran et al., 2016 [27]	RCT	EG: active tDCS + R-AAT CG: sham tDCS + R-AAT	EG: 4 CG: 4	EG: C3-C6, ASIA C, D CG: C3-C7 ASIA C, D	EG: 25.2 (10.4) mo; range = 7–48 mo GC: 141.2 (48.2) mo; range = 47–244 mo	EG: 49.7 (10.8) yrs; range = 36–62 yrs GC: 55.7 (5.9) yrs; range = 50–63 yrs	9
Potter-Baker et al., 2017 [28]	RCT	EG: active tDCS+ MP CG: sham tDCS+MP	EG: 4 CG: 4	EG: C2, C4-C6, AIS B, D CG: C3-C5, AIS B, D	EG: 54.5 (15.4) mo; range = 30–98 mo GC: 164 (76.8) mo; range = 21–372 mo	EG: 52 (1.6) yrs; range = 48–56 yrs CG: 55 (2.4) yrs; range = 51–62 yrs	5
Cortes et al., 2017 [29]	Crossover	EG: 1 mA active tDCS; 2 mA active tDCs CG: sham tDCS	11	C5-C7, AIS B-D	8.18 (5.74) yrs; range = 6–22 yrs;	44.9 (12.9) yrs; range = 21–63 yrs	6
Murray et al., 2015 [30]	Crossover	EG: 1 mA active tDCS; 2 mA active DCs CG: sham tDCS	9	C4-C6, AIS B, C	70.2 (32.4) mo; range = 9–126 mo	40.8 (14.2) yrs, range = 20–56 yrs	6
**TSCS (Transcutaneous Spinal Cord Stimulation)**							
Inanici et al., 2021 [31]	Crossover	EG: tSCS+ hand training CG: hand training	6	C3 and C5, AIS B-D	4.6 (3.8) yrs, range = 1.5–12 yrs	42.7 (14.2) yrs, range = 28–62 yrs	4
Garcia-Alén et al., 2023 [32]	RCT	EG: tSCS+R-AAT CG: R-AAT	21	EG: C3-C7, AIS A-D CG: C4-C7, T1, AIS A-D	EG: 5.5 (2.1) mo, range = 3–10 mo CG: 5.2 (2.2) mo, range = 2–9 mo	EG: 37.4 (13.3) yrs, range = 21–60 yrs CG: 38 (16.4) yrs, range = 18–70 yrs	5
**FES (Functional Electrical Stimulation)**							
Zoghi & Galea, 2018 [33]	Multicenter RCT	EG: FES + ReJoy CG: usual care	EG: 3 CG: 4	EG: C4, AIS A, D CG: C4, C6-C7, AIS B-D	EG: Unknowm CG: Unknown	EG: Unknowm CG: Unknown	1
Glinsky et al., 2009 [34]	RCT	EG: active FES + resistance training EC: sham FES + resistance training	EG: 32 CG: 32	EG: C4-C7, AIS complete and incomplete CG: C4-C7, AIS complete and incomplete	EG: range = 4–16 mo CG: range = 4–16 mo	EG: 38 (16) yrs, CG: 38 (16) yrs,	9
Kohlmeyer et al., 1996 [35]	RCT	EG: FES, Biofeedback, FES+ Biofeedback CG: conventional therapy	EG: FES: 10; Biofeedback: 13; FES+ Biofeedback: 11; CG: 10	FES: C4-C6, AIS complete and incomplete Biofeedback: C4-C6, AIS complete and incomplete FES+ Biofeedback: C4-C6, AIS complete and incomplete CG: C4-C6, AIS complete and incomplete	FES: 3.2 (0.9) weeks Biofeedback: 2.8 (0.8) weeks FES+ Biofeedback: 2.8 (0.8) weeks CG: 3.0 (0.9) weeks	FES: 32 (13) yrs Biofeedback: 38 (15) yrs FES+ Biofeedback: 42 (15) yrs CG: 43 (18) yrs	4
Popovic, 2006 [36]	RCT	EG: FES + repetitive grasping exercises CG: COT	EG: 12 CG: 9	EG: C4-C7, AIS A-D CG: C3-C7, AIS A-D	EG: 48.5 (38.2) days, range = 15–142 days CG: 76.2 (7.5) days, range = 15–243 days	EG: 34 (15.16) yrs, range = 16–65 yrs CG: 53.2 (13.6) yrs, range = 24–70 yrs	3
Kapadia, 2013 [37]	RCT	EG: FES+ functional patterns CG: COT	EG: 5 CG: 3	EG: C4-C6, AIS B; 3 of them don’t know CG: C4-C6, ASIA B	EG: not well specified CG: not well specified	EG: not well specified CG: not well specified	2
Popovic et al., 2011 [38]	RCT	EG: FES+COT CG: COT	EG: 9 CG: 12	EG: C4-C7, AIS B-D CG: C4-C6, AIS B-C	EG: 59.4 (31.8) days range = 33–134 days CG: 56.8 (24.7) days range = 22–102 days	EG: 41.5 (17.4) yrs range = 18–66 yrs CG: 44.9 (16.4) yrs range = 20–65 yrs	6
Kapadia et al., 2011 [39]	RCT	EG: FES+ ADL’s CG: COT	EG: 10 CG: 12	EG: C4-C7 CG: C4-C7	EG: 69.9 (14.11) days, range = 22–164 days CG: 58.33 (6.55) days, range = 22–102 days	EG: 43.2 (5.45) yrs CG: 44.75 (4.72) yrs	3
Kapadia, 2014 [40]	RCT	EG: FES+COT CG: COT1, COT2	EG: 10 CG: COT1 = 5, COT2 = 12	EG: C3-C6, AIS CG: COT1 = C3-C4, AIS COT2 = C4-C6, AIS	EG: 69.9 days CG: COT1 = 43.6 days, COT2 = 58.3 days	EG: 43.2 yrs CG: COT1 = 60.8 years, COT2 = 44.75 yrs	7
Harvey et al., 2017 [41]	Multicentre RCT	EG: FES + functional tasks (computer games) CG: usual care	EG: 37 CG: 33	EG: AIS A-D CG: AIS A-D	EG: 81 days, range = 45–110 days CG: 62 days, range = 45–110 days	EG: 81 yrs, range = 23–45 yrs CG: 29 yrs, range = 22–53 yrs	8
Anderson et al., 2022 [42]	Multicentre RCT	EG: FES + intensive task-specific hand-training program (computer games) CG: CT	EG: 27 CG: 24	EG: C4-C6, AIS B-D CG: C4-C7, AIS B-D	EG: 23.7 (12.9, 36.6) CG: 17.6 (7.4, 27.8)	EG: 40.0 (18.0), range = 22–58 yrs CG: 46.7 (17.2), range = 29–63 yrs	8
**TENS (Transcutaneous Electrical Nerve Stimulation)**							
Beekhuizen K. and Field-Fote E., 2005 [5]	RCT	EG: MP+TENS CG: MP	EG: 5 CG: 5	EG: C5-C7, AIS C, D CG: C5-C6, AIS C, D	EG: 29.6 (12.2) mo range = 12–43 mo GC: 58.6 (56.1) mo range = 12–154 mo	EG: 32.6 (8.0) yrs, range = 22–39 yrs; GC: 45 (10.3) yrs, range = 37–63 yrs	3
Beekhuizen K. and Field-Fote E., 2008 [43]	RCT	EG: MP+TENS, MP, TENS CG: continue their typical daily routines	EG: MP+TENS: 6; MP: 6; TENS: 6 CG: 6	MP+TENS: C5-C7, AIS C, D MP: C4-C7, AIS C, D TENS: C5-C7, AIS C, D GC: C5-C7, AIS C, D	MP+SS: 66.8 (97.1) mo range = 12–264 mo MP: 47.5 (52.9) mo range = 12–153 mo SS: 72.2 (47.3) mo range = 12–120 mo GC: 82.7 (78.8) mo range = 32–240 mo	MP+SS: 47.8 (20.0) yrs range = 22–70 yrs MP: 34.7 (14.9) yrs range = 21–64 yrs SS: 34.5 (14.9) yrs range = 19–56 yrs GC: 35.0 (6.8) yrs range = 24–41 yrs	4
Gomes- Osman et al., 2017 [44]	RCT	EG: FTP+TENS, TENS CG: conventional exercise training	EG: FTP+TENS: 14; TENS: 13 CG: 10	FTP+TENS: C5-C8, AIS B, C TENS: C4-C7, AIS A-D CG: C5-C7, AIS C, D	FTP+TENS: 13.7 (12.9) yrs TENS: 6.5 (9.0) yrs CG: 4.0 (3.8) yrs	FTP+TENS: 42.4 (13.5) yrs TENS: 34.2 (16.4) yrs CG: 36.6 (13.2) yrs	5
Nasser et al., 2014 [45]	RCT	EG: MP, MP+TENS CG: conventional rehabilitation	EG: MP: 10; MP+TENS: 10 CG: 5	MP: C5-C7, AIS C, D MP+TENS: C5-C7, AIS C, D CG: C5-C7, AIS C, D	MP: 21.8 (19.07) mo, range = 8–72 mo MP+TENS: 24.1 (22.07) mo, range = 6.84 mo CG: 18 (12.19) mo, range = 7–36 mo	MP: 33.2 (6.14) yrs, range = 25–45 yrs MP+TENS: 38.7 (12.09) yrs, range = 24–60 yrs CG: 33.4 (7.09) yrs, range = 25–41 yrs	5
**Neuromuscular Stimulation (NMS)**							
Needham-Shropshire et al., 1997 [46]	RCT	EG: NMS+ergometry, 4 weeks of NMS+ergometry & 4 weeks of ergometry CG: ergometry	EG: NMS+ergometry: 12; 4 weeks of NMS+ergometry & 4 weeks of ergometry: 11 CG: Ergometry: 11	Not avaible	EG: NMS+ergometry: 6 yrs 4 weeks of NMS+ergometry & 4 weeks of ergometry: 9 yrs Ergometry: 4 yrs	EG: NMS+ergometry: 24 yrs 4 weeks of NMS+ergometry & 4 weeks of ergometry: 22 yrs Ergometry: 24 yrs	2

Maximal score of the PEDro: 10 points (0 = worse, 10 = excellent). TMS: Transcranial Magnetic Stimulation; EG: Experimental Group; CG: Control Group; SD: Standard Deviation; mo: months; yrs: years; RCT: Randomized Controlled Trial; tDCS: Transcranial Direct Current Stimulation; R-AAT: Robot-Assisted Arm Training; MP: Massive Practice; tSCS: Transcutaneous Spinal Cord Simulation; FES: Functional Electrical Stimulation; COT: Conventional Occupational Therapy; TENS: Tanscutaneous Electrical Nerve Simulation; FTP: Functional Task Practice; NMS: Neuromuscular Stimulation.

**Table 2 sensors-24-04695-t002:** Intervention characteristics of the studies included in the systematic review.

Study	Intervention Duration	Frequency Sessions	Duration of Each Session	Type of Interventions	Functionality and Motor Function Outcome Measurements
**TMS (Transcranial Magnetic Stimulation)**					
Gharooni et al., 2018 [23]	2 weeks, 10 sessions	5x/week	Not available	**Active iTBS:** coil over M1 of hand. 3 stimuli at 50 Hz repeated at 200 ms intervals for 2 s. Intertrain interval of 8 s, repeated 20 times for a total of 600 pulses in 200 s. Intensity at 80% RMT **Sham iTBS:** coil rotated 90° about its vertical midline.	UEMS SCIM
Kuppuswamy et al., 2011 [24]	1 week, 5 sessions	5x/week	15 min	**Active rTMS**: coil over the lowest threshold spot for eliciting a MEP in FDI, thenar eminence or ECR. 5 Hz as 2 trains separated by 8 s for 15 min. Intensity at 80% of the AMT. **Sham rTMS**: 5% of real stimulator output.	UEMS ARAT NHPT
Gomes-Osman and Field-Fote, 2015 [25]	3 days, 3 sessions	Not available	Not available	**Active rTMS:** coil over thenar muscles, hemisphere contralateral to the weaker hand. 10 Hz, 800 pulses distributed in 2 s trains of 40 pulses, inter-train interval of 30 s during subjects practiced a fine motor task. Intensity at 80% of biceps RMT. **Sham rTMS:** using a previously validated approach that mimics the experience of the real rTMS	JTHFT Pinch strength Grasp strength
Belci et al., 2004 [26]	1 week, 5 sessions	5x/week	1 h	**Active rTMS:** coil over the left motor cortex. 0.1 Hz, double pulses separated by 100 ms (10 Hz), 10 s interval. Intensity 90% MEPs hand muscles. **Sham rTMS:** coil over the occipital cortex, 360 doublet pulses	AIS NHPT
**tDCS (Transcraneal Direct Currrent Stimulation)**					
Yozbatiran et al., 2016 [27]	10 sessions	Unknown	20 min tDCS 60 min MAHI-Exo II trainig	**Active tDCS:** anode on C3/C4 contralateral to the targeted arm, cathode over contralateral supraorbital area. 20 min, 2 mA anodal direct current. **Sham tDCS:** first 30 s the current was ramped up to 2 mA and during last 30 s ramped down. **MAHI-Exo II**: repetitive movement training.	JTHFT MAL UEMS
Potter-Baker et al., 2018 [28]	2 weeks, 10 sessions	5x/week	2 h	**Active tDCS:** anode over M1 of more weaker muscle of upper limb, cathode over the contralateral supraorbital region. 2 mA during the first 30 min of the first hour of MP training + the first 30 min of the second hour of MP training **Sham tDCS:** “sham setting”. **MP:** training program was individualized based on their deficit.	MMT UEMS ARAT NHPT
Cortes et al., 2017 [29]	1 session	Unknown	20 min	**Active tDCS:** anode onC3/C4 contralateral to the test hand, cathode over the contralateral supraorbital area. 20 min, 1° 2 mA. **Sham tDCS:** 30 s ramp up at the beginning and ramp down at the end of the stimulation.	Hand robot evaluation: mean velocity, peak velocity, smoothness, and duration of the movement BBT
Murray et al., 2015 [30]	1 session	Unknown -	20 min	**Active tDCS:** anode over M1 (right ECR), cathode contralateral supraorbital area. 20 min, 1° 2 mA. **Sham tDCS**: 20 min, short ramp up/down event at the beginning and end of the stimulation period without any current between the 2 events.	MVC
**tSCS (Transcutaneous Spinal Cord Stimulation)**					
Inanici et al., 2021 [31]	1 month, 12 sessions	3x/week	2 h	**tcSCS:** 2 cathodes at level of the lesion and below, 2 anodes at anterior superior iliac spine. Biphasic or monophasic, 1 ms pulse, 30 Hz (10 KHz), 40–90 mA, intensity adjusted on subjects feedback. **Intensive functional task training:** 1–2 exercises of each category: unimanual and bimanual activities of gross upper-limb movements, isolated finger movements, bimanual task performance and simple and complex pinch.	GRASSP UEMS Lateral pinch force SCIM
García-Alén et al., 2023 [32]	2 weeks, 8 sessions	4x/week	1 h	**tSCS:** cathodes at C3-C4 and C6-C7, 2 anodes at anterior superior iliac spine. Biphasic, 1 ms pulse, 30 Hz, at intensity of 90% rest motor threshold of APB. **Armeo Power:** 6 exercises for each upper limb: 4 exercises for f open/close hand, 2 exercises for reaching and grasping.	MVC (cylindrical grasp, lateral and tip to tip pinch) UEMS GRASSP SCIM
**FES (Functional Electrical Stimulation)**					
Zoghi & Galea, 2018 [33]	8 weeks, 40 sessions	5x/week	Unknown	**FES:** forearm and wrist muscles. Biphasic, 200 μs pulse, 50 Hz. **ReJoyce:** hand tasks (reaching, grasping, manipulating, pulling, rotation and releasing).	ARAT GRASSP
Glinsky et al., 2009 [34]	8 weeks, 24 sessions	3x/week	Unknown	**Active FES:** wrist muscles. 6:6 s on/off ratio, 0.3 ms pulse, 50 Hz, 70 mA or the maximum intensity tolerated. **Sham FES:** radial and ulnar styloid process. 1 Hz, 6:6 s on/off ratio, 1 mA. **Resistance training:** 6 sets of 10 repetitions (wrist extension or wrist flexion) with 1–3 min rest between sets. Load was initially set prior to training using one set of 10 repetitions and increased according to the principles of progressive resistance training.	MVC
Kohlmeyer et al., 1996 [35]	5–6 weeks, 25–30 sessions	5x/week	20–40 min	**FES:** wrist extensors. 8:8 s on/off ratio and ramp up/down times 2 s, 0.3 ms pulse, cyclic stimulation, 20 Hz, intensity adjusted to an appropriate level or tolerance. **Biofeedback:** observe the EMG of their wrist extensors on a video display screen and listen to audio feedback, while subjects attempt to active their wrist extensors. **Conventional treatment:** passive range of motion, orthotic intervention, strengthening, functional activities.	MMT Function score evaluation (evaluation of four graded self feeding abilities)
Popovic, 2006 [36]	12 weeks, 60 sessions	5x/week	45 min	**FES:** muscles could be stimulated using surface FES technology and which combination of muscle contractions generated the palmar and/or the lateral grasp. Balanced, Biphasic, 250 μs pulse, 20–70 Hz, 8–50 mA. **COT:** muscle facilitation exercises, task specific, repetitive functional training, strengthening and motor control training, stretching exercises, ADLs, caregiver training.	FIM SCIM REL test
Kapadia, 2013 [37]	13–16 weeks, 39 sessions	3x/week	1 h	**FES:** FCR, FCU, ECR, ECU, FD, ED, thumb abductors, thumb flexors, thumb oppositors. Biphasic, 250 μs pulse, 40 Hz, 8–50 mA. **COT:** muscle facilitation exercises, task specific, repetitive functional training, strengthening and motor control training, stretching exercise, electromuscular stimulation, ADLs, caregiver training.	TRI-HFT GRASSP FIM SCIM
Popovic et al., 2011 [38]	8 weeks, 40 sessions	5x/week	2 h	**FES:** FCR, FCU, ECR, ECU, FD, ED, ECU, thumb abductors, thumb flexors, thumb oppositors. Balanced, biphasic, 250 μs pulse, 40 Hz, 8–50 mA. **ADL’S** **COT:** muscle facilitation exercises, task specific, repetitive functional training, strengthening and motor control training, stretching exercises, electrical stimulation for muscle strengthening, ADLs, caregiver training.	FIM SCIM TRI-HFT
Kapadia et al., 2011 [39]	8 weeks, 40 sessions	5x/week	1 h	**FES:** ADL’s + FES. Balanced, biphasic, 250 μs pulse, 40 Hz, 8–50 mA. **COT:** strengthening and stretching exercises, ADLs.	FIM SCIM TRI-HFT
Kapadia, 2014 [47]	COT 1: 12 weeks, 60 sessions COT 2: 8 weeks, 80 sessions COT+FES: 8 weeks, 40 sessions	COT 1: 5x/week COT 2: 2 times per day, 5x/week COT+FES: 5x/week	COT 1: 1 h COT 2: 2 h COT+FES: 1 h COT + 1 h FES	**FES**: ECR, ECU, FCR, FCU, FD, ED, thumb abdcutors, thumb flexors, thumb oppositors. Balanced, biphasic, 250 μs pulse, 40 Hz, 8–50 mA. ADLs + FES. **COT:** strengthening and stretching exercises, ADLs, muscle facilitation exercises, task specific, repetitive functional training, electrical stimulation and caregiver training.	FIM SCIM
Harvey et al., 2017 [41]	8 weeks, 40 sessions	5x/week	1 h	**FES:** any or all of the muscles that facilitate opening or closing hand. Biphasic, 200 μs pulse, 50 Hz. **Intensive task-specific hand-training program:** reaching, grasping, manipulating, pulling, rotating and releasing (computer games). **Usual care:** physiotherapy, vocational, recreational and occupational therapy.	ARAT GRASSP CUE SCIM
Anderson et al., 2022 [42]	14 weeks, 36–40 sessions	3–5x/week	1 h	**FES:** movement’s patterns (palmar grasp, lateral pinch grasp, pinch grasp, lumbrical grasp, tripod grasp, side reach with finger extension, forward reach and grasp, and hand to mouth). Parameters were selected from pre-programmed stimulation protocols. **CT:** reach or prehension movements, bilateral task-specific movements, range of motion and mobilization of joints, splinting, sensorimotor stimulation, electrical stimulation, and reduction of edema.	SCIM III TRI-HFT GRASPP
**TENS (Transcutaneous Electrical Nerve Stimulation)**					
Beekhuizen K. and Field-Fote E., 2005 [5]	3 weeks, 15 sessions	5x/week	2 h	**TENS:** anode at wrist, cathode at 2 cm. 1 Hz, each train consists of 5 single pulses, 1 ms, 10 Hz, intensity compound muscle action potentials from the APB. **MP:** repetition of tasks in each of 5 categories: gross upper-limb movement, grip, grip with rotation, pinch and pinch with rotation. Each category has 10 tasks, 25 min in each the next category.	Pinch grip force WMFT JHFT
Beekhuizen K. and Field-Fote E., 2008 [43]	3 weeks, 15 sessions	5x/week	2 h	**TENS:** anode at wrist, cathode at 2 cm. Trains of electric stimulation (10 Hz; on/off duty cycle, 500/500 ms; 1 ms pulse) at 1 Hz. Intensity adjusted to elicit a visible twitch of the thumb muscles, reduced it to a level at which no visible twitch was observed. **MP:** repetitive practice of tasks in each of 5 categories: gross upper-limb movement, grip, grip with rotation, pinch and pinch with rotation. Each category had 14 specific tasks, 25 min before moving on to the next category.	JTHFT WMFT Pinch grip force
Gomes- Osman et al., 2017 [44]	4 weeks, at least 17 sessions	5x/week	2 h	**TENS**: bilaterally, electrodes placed on the volar aspect of each wrist targeting the median nerve. 10 Hz, 1 ms pulse duration, on/off duty cycle 500 ms/500 ms. Stimulation intensity was increased to an intensity at which a muscle twitch could be observed in the thumb, and then decreased below this level for the remainder of the session. **FTP:** practice 6 categories of bimanual activities, 20 min each category (independent finger movement, precision grip, pinch with object manipulation, power grip, complex power grip, finger isolation, whole arm movement).	Pinch force Cylindrical grasp force
Nasser et al., 2014 [45]	3 weeks, 15 sessions	5x/week	2 h	**TENS:** anode at the wrist, cathode 2 cm proximal to it. 1 Hz, each train consisted of 5 single pulses at 1 ms duration delivered at 10 Hz with stimulus intensity just below that which evoked an observable twitch in any of the muscles innervated by the median nerve. **MP**: repetition of tasks in each of 5 categories (gross upper-limb movement, grip, grip with rotation, pinch, pinch with rotation). Performed the tasks within each category for 25 min.	EMS Pinch grip force WMFT JTHFT
**NMS (Neuromuscular Stimulation)**					
Needham-Shropshire et al., 1997 [46]	8 weeks, 24 sessions	3x/week	Unknown	**NMS:** proximal electrode near motor point of triceps, other at 5 cm, 250 μs pulse, 50 Hz, intensity adjusted at optimal contraction of triceps, during 10 min. **Arm ergometer exercise**: 4 or 5 min exercise intervals of cranking with 3 min rest periods between intervals. Speed of the flywheel at 60 RPM for each period of exercise. Resistance adjusted for each subject.	Manual muscle test (ASIA motor): Biceps, triceps, wrist flexors and extensors

TMS: Trancranial Magnetic Stimulation; iTBS: Theta-Burst Stimulation; RMT: Rest Motor Threshold; UEMS: Upper Extremity Motor Score; SCIM: Spinal Cord Independence Measure; rTMS: Repetitive Transcranial Magnetic Stimulation; RTP: Repetitive Task Practice; MEP: Motor Evoked Potential; FDI: First Dorsal Interosseous; ECR: Extensor Carpo Radial; AMT: Active Motor Threshold; ARAT: Action Research Arm Test; NHPT: Nine Hole Pegboard Test; tDCS; Transcranial Direct Current Stimulation; JTHFT: Jebson Taylor Hand Function Test; MAL: Motor Activity Log; MP: Massive Practice; MMT: Manual Muscle Test; BBT: Box and Block Test; MVC: Maximal Voluntary Contraction; tSCS: Transcutaneous Spinal Cord Stimulation; GRASSP: Graded Redefined Assessment of Strength, Sensibility, and Prehension; APB: Abductor Pollicis Brevis; FES: Functional Electrical Stimulation; BMCA: Brain Motor Control Assessment; EMG: Electromyography; ADLs: Activities of Daily Living; FIM: Functional Independence Measure; CT: Conventional Therapy; SCI-QOL: Spinal Cord Injury–Quality of Life.

## Data Availability

Not applicable.

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
