# Peer review of "Noninvasive Electromagnetic Neuromodulation of the Central and Peripheral Nervous System for Upper-Limb Motor Strength and Functionality in Individuals with Cervical Spinal Cord Injury: A Systematic Review and Meta-Analysis"

_sensors, 2024, doi:10.3390/s24144695_

Round 1

Reviewer 1 Report

Comments and Suggestions for Authors

In this manuscript, the authors provide a comprehensive overview of the current state of noninvasive electromagnetic neuromodulation techniques applied to individuals with cervical spinal cord injury (cSCI). The authors systematically reviewed various neuromodulation approaches, including transcranial magnetic stimulation (TMS), transcranial direct current stimulation (tDCS), transcutaneous spinal cord stimulation (tSCS), functional electrical stimulation (FES), transcutaneous electrical nerve stimulation (TENS), and neuromuscular stimulation (NMS). The primary focus was on the restoration of upper limb functionality and motor function. The systematic review and meta-analysis approach is methodologically sound and provides valuable insights into the efficacy of noninvasive neuromodulation techniques. Therefore, this work is meaningful and valuable and can be accepted with minor revisions, with the following detailed comments:

  1. Could the authors provide more details on the study selection and inclusion criteria? For instance, were factors such as study design, sample size, or follow-up duration considered?
  2. The review includes 25 studies. Could the authors elaborate on the process of data extraction and how data from different studies were synthesized?
  3. Could the authors provide more insight into the reasons behind the heterogeneity of the evaluation methods and how this might impact the generalizability of the findings?
  4. How did the authors ensure the quality and relevance of the studies included in the review, given the broad range of neuromodulation techniques and patient populations?
  5. How did the authors assess the quality of the included studies and control for potential bias? Were tools such as the PEDro scale used to evaluate the scientific rigor of the studies?
  6. What specific recommendations can the authors provide for future studies to improve the standardization of outcome measures and enhance the possibility of performing a meta-analysis? In addition, Given the limitations of current research, could the authors propose directions for future research to further explore the application of non-invasive electromagnetic neuromodulation in the rehabilitation of cSCI patients?
  7. The paper suggests that non-invasive electromagnetic neuromodulation combined with rehabilitation significantly improves upper limb functionality in cSCI subjects. Could the authors further discuss the clinical implications of these findings and address potential limitations and biases?
Comments on the Quality of English Language

the english is well.

Author Response

Thank you very much for taking the time to review this manuscript. Please find the detailed responses below and the corresponding corrections in the re-submitted files.

Comment 1:Could the authors provide more details on the study selection and inclusion criteria? For instance, were factors such as study design, sample size, or follow-up duration considered?Thank you for pointing this out. Done as suggested: In page 3, headland “2.2 study selection procedure” we specified lines 132-133 lines: “(iv) randomized controlled clinical trials or those with a crossover design”. The simple size or follow-up duration was not considered, as exclusion criteria.

Comment 2:The review includes 25 studies. Could the authors elaborate on the process of data extraction and how data from different studies were synthesized?We added more information in this point as commented by reviewer. In the text, page 4, headland 2.4. “data extraction and statistical analysis” we specified 150-165 line: “all the data was extracted from the studies included by four different authors. The following descriptive data was extracted: authors, publication year, study design, number of subjects in each experimental group, subject clinical and demographic characteristics, intervention characteristics (type, frequency and duration of the sessions, stimulation parameters and total duration of the intervention), and outcome measures related to the functionality and motor function of the upper extremities”.

Comment 3: Could the authors provide more insight into the reasons behind the heterogeneity of the evaluation methods and how this might impact the generalizability of the findings?We added more insight in this point. In the text, page 17, headline “4.discussion”, line 438 have added: Also, there was big variability in the evaluation of muscle strength and/or functionally of UE in SCI and the expression of those data; for example one study gave only % changes without any rough data or mean+/- SD (Beekhuizen K. et al., 2005).  There should be general concept for clinical assessments for UE at least such as UEMS of ASIA for muscle strength and GRASSP for functionally assessment and the data should be expressed at least as rough data and group mean.”

Comment 4:How did the authors ensure the quality and relevance of the studies included in the review, given the broad range of neuromodulation techniques and patient populations?Thank you for pointing this out. One of the aims of this study was to study the quality of the neuromodulation studies for UE according to each type of neuromodulation technic. And we used PeDro Scale to study the quality and relevance of the studies in page 3 point 2.3.

Comment 5:How did the authors assess the quality of the included studies and control for potential bias? Were tools such as the PEDro scale used to evaluate the scientific rigor of the studies?PEDro scale was used to evaluate the scientific rigor of the studies; it is specified in headline 2.3 assessment of methodological quality, page 3. Also we specified for each technique, in headline “3. Results” if they applied blinded interventions, standardized protocols, missing data and outcome measures used, theses factors control for potential bias.

Comment 6:What specific recommendations can the authors provide for future studies to improve the standardization of outcome measures and enhance the possibility of performing a meta-analysis? In addition, given the limitations of current research, could the authors propose directions for future research to further explore the application of non-invasive electromagnetic neuromodulation in the rehabilitation of cSCI patients? We agree with this comment. In the text in page 18, headline 5.  Conclusion, we have added: “In order to perform a meta-analysis, more randomized controlled trials with standardized outcome measures for the UL in cSCI are needed. Future research should prioritize the use of standardized outcome measures to facilitate meta-analysis. Encoring authors to adopt commonly accepted scales at least such as UEM for UL motor function and GRASSP for UL functionality in cSCI. The other clinical assessment could be done additionally such as UEMS ARAT and NHPT and the data should be give adequately such as rough data, mean and SD”.

Comment 7: The paper suggests that non-invasive electromagnetic neuromodulation combined with rehabilitation significantly improves upper limb functionality in cSCI subjects. Could the authors further discuss the clinical implications of these findings and address potential limitations and biases? We are so sorry we made a mistake

 but we have changed. Page 17, headline “4. Discussion”, line 436: “This systematic review investigated the effects of different noninvasive electromagnetic neuromodulation techniques combined with or without rehabilitation for improving the motor function and functionality of ULs in subjects with cSCI”. In page 18, line 481: “Two studies showed improvements in UL motor function and functionality, and pinch force when applied TENS alone, but the results where higher significant when was combined with rehabilitation (44,46)”. Finally, we have added in text, page 18, headline “5. Conclusions”: “Future research should prioritize the use of standardized outcome measures to facilitate meta-analysis. Encoring authors to adopt commonly accepted scales such as UEM for UL motor function and GRASSP for UL functionality in cSCI. The other clinical assessment could be done additionally such as UEMS ARAT and NHPT and the data should be give adequately such as rough data, mean and SD. Some studies have low methodological quality, as indicated by low PEDro scores. Future researches should emphasize the importance of high-quality study designs, such as randomized controlled trials, with clear reporting and adherence to established methodological guidelines like the CONSORT statement”.

Reviewer 2 Report

Comments and Suggestions for Authors

My comments are in the attached pdf file.

Author Response

Thank you very much for taking the time to review this manuscript. Please find the detailed responses below and the corresponding corrections in the re-submitted files.

Comment 1: Lack of Meta-Analysis Completion.The meta-analysis could not be completed due to the lack of common motor or functional evaluations across studies. Future research should prioritize the use of standardized outcome measures to facilitate meta-analysis. Encouraging authors to adopt commonly accepted scales for upper limb functionality and motor function, such as ARAT, NHPT, and UEMS, would be beneficial. We agree with this comment. Therefore, we have added in page 17, headline “4. Discussion”, line 438:  “Unfortunately, the meta-analysis could not be completed due to a lack of common motor or functional evaluations across studies. Also, there was big variability in the evaluation of muscle strength and/or functionally of UE in SCI and the expression of those data; for example one study gave only % changes without any rough data or mean+/- SD (Beekhuizen K. et al., 2005). There should be general concept for clinical assessments for UE such as UEMS of ASIA for muscle strength and GRASSP for functionally assessment and the data should be expressed as rough data and group mean.In page 18, headline “5. Conclusions”, line 499: “Future research should prioritize the use of standardized outcome measures to facilitate meta-analysis. Encoring authors to adopt commonly accepted scales such as UEMSfor UL motor function and GRASSP for UL functionality in cSCI. The other clinical assessment could be done additionally such as UEMS ARAT and NHPT and the data should be give adequately such as rough data, mean and SD”.

Comment 2: Heterogeneous Sample Characteristics. The included studies have highly heterogeneous samples regarding the severity of SCI, time since injury, and age, impacting the power and generalizability of the results. Conducting subgroup analyses or stratified meta-analyses based on these variables can provide more nuanced insights. Future studies should aim for more homogeneous sample populations or explicitly address and adjust for these variabilities in their analyses. We agree with this comment. Therefore we have added in page 17, headline “4. Discussion”, line 447: “Also, the samples were highly heterogeneous regarding the severity of SCI, time since injury, and age, impacting the power and generalizability of the results. Conduction subgroup analyses or stratified meta-analysis based on theses variables can provide nuanced insights. Future studies should aim for more homogenous sample population or explicitly address and adjust for these viabilities in their analyses”.

Comment 3: Insufficient Detail on Adverse Effects. The reporting of adverse effects is inconsistent, with some studies not mentioning them at all. Mandate comprehensive reporting of adverse effects in future studies to better understand the safety profile of each neuromodulation technique. Including a standardized section on adverse effects in study reports could be helpful. We agree with this comment. Therefor we have added in page 17, headline “4. Discussion”, line 469: “The reporting of adverse effects is inconsistent, with some studies not mentioning them at all. A comprehensive reporting of adverse effects in future studies would be important to better understand the safety profile of each neuromodulation technique.

Comment 4: Small Sample Sizes. Many studies included in the review have small sample sizes, limiting the statistical power and reliability of the findings. Encourage multicenter trials and collaboration to increase sample sizes. Providing guidelines for the minimum sample size required to achieve adequate power in neuromodulation studies can also be beneficial. We agree with this comment. Therefore we have added in page 17, headline “4. Discussion”, line 447: “Except 4 studies (31,33,35,42),the sample sizes were small, oscillating from 3 to 15 subjects in each experimental group, limiting the statistical power and reliability of the findings. It is necessary encourage multicenter trials and collaboration to increase sample sizes.

It is very difficult to give any samples size when there are may neuromodulation techniques. A pilot study should be conducted to obtain reliable estimates of effect size and variability.

Comment 5: Limited Long-term Follow-up Data.The review lacks information on the long-term efficacy and sustainability of the improvements seen with neuromodulation techniques. Future studies should include longer follow-up periods to assess the durability of the therapeutic effects. Including long-term follow-up data will provide a better understanding of the sustained benefits and potential late-onset adverse effects. We agree with this comment. Therefore we have added in page 18, headline “4. Discussion”, line 481: “Future studies should include longer follow-up periods to assess the durability of the therapeutic effects”

Comment 6: Inconsistent Reporting of Intervention Parameters. There is variability in how intervention parameters such as intensity, frequency, and duration are reported. Adopt a standardized reporting template for intervention parameters to ensure consistency and facilitate replication. Detailed documentation of these parameters should be a requirement for publication. Thank you for pointing this out. Because of lack of literature is difficult to make conclusion and suggest specific intervention parameters to ensure consistency and facilitate replication.

Comment 7: Low Methodological Quality in Some Studies. Some studies have low methodological quality, as indicated by low PEDro scores. Emphasize the importance of high-quality study designs, such as randomized controlled trials, with clear reporting and adherence to established methodological guidelines like the CONSORT statement. Providing training and resources on high-quality research methods to investigators in this field may help improve the overall quality of studies. We agree with this comment. Therefore we have added in page 18, headline “5. Conclusions”: “Some studies have low methodological quality, as indicated by low PEDro scores. Future researches should emphasize the importance of high-quality study designs, such as randomized controlled trials, with clear reporting and adherence to established methodological guidelines like the CONSORT statement”.
